# The Physiological and Pathological Roles of Mitochondrial Calcium Uptake in Heart

**DOI:** 10.3390/ijms21207689

**Published:** 2020-10-17

**Authors:** Lo Lai, Hongyu Qiu

**Affiliations:** Center for Molecular and Translational Medicine, Institute of Biomedical Science, Georgia State University, 100 Piedmont Ave, PSC 588, Atlanta, GA 30303, USA; llai@gsu.edu

**Keywords:** mitochondria, Ca^2+^ uptake, cardiac function, heart failure, valosin-containing protein

## Abstract

Calcium ion (Ca^2+^) plays a critical role in the cardiac mitochondria function. Ca^2+^ entering the mitochondria is necessary for ATP production and the contractile activity of cardiomyocytes. However, excessive Ca^2+^ in the mitochondria results in mitochondrial dysfunction and cell death. Mitochondria maintain Ca^2+^ homeostasis in normal cardiomyocytes through a comprehensive regulatory mechanism by controlling the uptake and release of Ca^2+^ in response to the cellular demand. Understanding the mechanism of modulating mitochondrial Ca^2+^ homeostasis in the cardiomyocyte could bring new insights into the pathogenesis of cardiac disease and help developing the strategy to prevent the heart from damage at an early stage. In this review, we summarized the latest findings in the studies on the cardiac mitochondrial Ca^2+^ homeostasis, focusing on the regulation of mitochondrial calcium uptake, which acts as a double-edged sword in the cardiac function. Specifically, we discussed the dual roles of mitochondrial Ca^2+^ in mitochondrial activity and the impact on cardiac function, the molecular basis and regulatory mechanisms, and the potential future research interest.

## 1. Introduction

The heart is one of the highest energy-demanding organs as it beats continuously throughout the entire course of a lifetime. Mitochondria are the powerhouses in cells that re-supply the cells with adenosine triphosphate (ATP) and provide the most energy to the heart [1]. It is well known that Ca^2+^ acts as an essential regulator participating in multiple stages of the bioenergetic procedures, particularly in the initialization and activation of mitochondrial respiration and ATP generation. A deficiency of cardiac mitochondrial Ca^2+^ would impair the mitochondrial function and reduce the energy supply, leading to cell damage or death. Besides the energy supply, mitochondria Ca^2+^ homeostasis is also essential in other biological functions in cardiomyocytes. For example, mitochondria could buffer the concentration of cytosolic Ca^2+^, which stimulates cardiac excitation-contraction coupling (ECC) [2,3]. When this function of mitochondria is impaired, cardiac contractility would be interfered, leading to cardiac dysfunction [2].

Reciprocally, under the pathological conditions, such as ischemia, infarction, and pressure overload, stress induces excessive calcium accumulation in the cardiomyocytes, resulting in mitochondrial Ca^2+^ overload. Studies have shown that excessive mitochondrial Ca^2+^ results in an increase in the generation of reactive oxygen species (ROS), impairment of mitochondrial membrane potential (ΔΨ), and mitochondrial permeability transition pore (mPTP) opening as well as ATP depletion, which subsequently leads to cell death and cardiac dysfunction [4,5]. These results indicate that mitochondrial Ca^2+^ plays a dual role in mediating cardiac function, e.g., at the physiological ranges, the increase of mitochondrial Ca^2+^ will enhance energy production and cardiac contractility. In contrast, excessive mitochondrial Ca^2+^ will induce the deleterious effects leading to cell damage and functional impairment. Therefore, maintaining the mitochondrial Ca^2+^ homeostasis is crucial for cardiomyocyte survival and function.

Despite the importance of mitochondrial Ca^2+^ homeostasis, the underlying regulatory mechanisms are far from fully understood. In general, the physiological concentration of mitochondrial [Ca^2+^] is constantly controlled through two major operative regulatory systems by releasing Ca^2+^ into the cytosol or up-taking Ca^2+^ from the cytosol. Voltage-dependent anion channel 1 (VDAC1), located at the outer mitochondrial member, is the first gatekeeper of Ca^2+^ entering mitochondria [6]. Ca^2+^ was kept at the mitochondrial intermembrane space (IMS). Then, it enters the mitochondrial matrix through mitochondrial Ca^2+^ uniporter (MCU). By interacting with a variant of regulatory proteins, such as calcium uptake proteins (MICUs), MCU constitutes a comprehensive functional complex to control the uptake of Ca^2+^ into the mitochondrial matrix. On the other hand, Ca^2+^ stored in the matrix can be pumped back to IMS through Na^+^/Ca^2+^ (NCX) or H^+^/Ca^2+^ exchangers [7], as well as by mPTP opening when the cell is under stress [8,9] (Figure 1). Although the uptake and release of the mitochondrial Ca^2+^ appear as two independent systems, recent studies indicate that an interlink may exist between the two systems. The uptake of mitochondrial Ca^2+^ may affect the initiation and activation of the releasing system, particularly mPTP opening under the stress conditions [10,11,12]. In addition to the conventional regulatory apparatus, more new regulators are being discovered to be involved in regulating mitochondrial Ca^2+^ homeostasis and participate in the cardiac mitochondrial function, such as valosin-containing protein (VCP), a new identified ATPase, which was found to be involved in both mitochondrial calcium uptake and mPTP opening [13]. A better understanding of the underlying mechanism would bring new insights into the molecular mechanism underlying human heart disease and lead to the discovery of new targets for the development of therapeutic strategies.

In this review, we summarized the recent findings in the studies on the regulation of Ca^2+^ uptake in terms of its dual roles in mitochondrial function and the impact on the cardiac diseases, the molecular basis and regulatory signaling, and the novel regulators, providing the updated information and potential future research direction in this field.

## 2. The Dual Roles of Mitochondrial Ca^2+^ in Regulating Cardiac Function

As discussed above, mitochondrial Ca^2+^ plays a dual role in regulating cardiac function. In physiological conditions, Ca^2+^ in mitochondria can activate ATPase activity in as short a time as one minute [15] and as low as nanomolar concentration [16]. When energy demand is increased due to the requirement of cardiac workloads, such as under the exercise condition, mitochondrial Ca^2+^ would be increased to activate dehydrogenase activity and enhance the oxidation rate of nicotinamide adenine dinucleotide hydrogen (NADH), increasing ATP production [4]. It has been shown that mitochondrial Ca^2+^ could regulate ATPase and ATPase inhibitors to fit the change of cardiac energy demand [17,18,19].

Moreover, in the cardiomyocytes, Ca^2+^ is released from the sarcoplasmic reticulum (SR) and binds to troponin C, resulting in an interaction between myosin and actin filaments, stimulating ECC leading to cardiac contractions. Mitochondria participate in this process of ECC by buffering the cytosolic Ca^2+^ and storing Ca^2+^ in the matrix to generate ATP for the energy required for contraction [4]. During a heavy workload, as stimulated through the β-adrenergic receptor, the matrix Ca^2+^ concentration could be increased dramatically to 500 to 800 nmol/L compared to 100–200 nmol/L at a normal pace (measured in isolated mitochondria) [20], to meet the high energy requirement. Thus, any reason causing the deficiency of mitochondrial Ca^2+^ upload would impair the heart’s energy production and contractile capability.

On the contrary, excessive cardiac mitochondrial Ca^2+^ results in oxidative stress and leads to cell death. Cardiac mitochondria produce ROS throughout the respiration chain. Ca^2+^ promotes ATP production and speeds up the ROS accumulation as well. Excessive ROS induces cardiomyocyte injury, leading to the development of cardiovascular diseases, like ischemia/reperfusion (I/R), myocardium infraction, pressure overload-induced cardiomyopathy and chronic heart failure [12,21]. ROS can lead to mitochondrial dysfunction and cell death by directly inducing the structure change of mPTP complex, resulting in pore opening as found in I/R or pressure overload-induced heart failure [21]. ROS/Ca^2+^ can also lead to cardiac dysfunction through other molecules. For example, mitochondrial calmodulin-dependent protein kinase II (CaMKII), a protein found in the mitochondrial matrix, can be activated by both ROS and Ca^2+^, involving in the mPTP opening under chronic catecholamine stress [22], and promotes the progress of heart failure [23].

The impact of mitochondrial Ca^2+^ in chronic heart failure was extensively investigated (as described in Raffaello’s review) [12,24]. The dysregulated Ca^2+^ handling was found to be highly associated with heart failure [25]. Numerous studies in animal models have demonstrated that alterations of intracellular calcium are primarily responsible for the depressed contractility of the failing heart, which have been well-reviewed previously [9,26,27]. The majority of data demonstrating Ca^2+^ toxicity in mitochondria are derived from experiments manipulating intracellular Ca^2+^ homeostasis using genetic approaches. The increasing evidence from these animal models indicated that cytosolic Ca^2+^ overload induced the impairment of mitochondrial calcium hemostasis, resulting in the opening of mPTP, increased mitochondrial oxidative stress, collapse of mitochondrial membrane potential, impaired ATP production, and necrosis of cardiomyocytes, subsequently leading to heart failure [9].

Three significant mechanisms were revealed to be involved in the Ca^2+^-induced contractile dysfunction. The first mechanism consists of downregulation of SR Ca-ATPase and upregulation of the NCX function, both tend to shift Ca^2+^ out of the cell and reduce SR Ca content. This reduces the SR Ca available to be released during subsequent ECC [28]. The second mechanism relates to the increase in cytosolic Na^+^ levels. It causes mitochondrial Ca^2+^ transport out from the matrix through NCX, and impaired Krebs cycles lead to an imbalance between the supply and demand of the energy for the workload [29]. The third mechanism is mediated by MCU’s inactivation, which leads to impaired ATP production and ROS accumulation [30]. Although the exact mechanisms remain mostly unknown, emerging evidence indicates an acceptable concept that dysregulation of Ca^2+^ in mitochondria could impact on cardiomyocyte survival and contractility through two primary mechanisms: the reduction of ATP production and the increase of mPTP opening.

## 3. Mitochondrial Ca^2+^ Uptake, a Double-Edged Sword for Cardiac Mitochondrial Functions

ATP generation and mPTP opening are two major events occurring in mitochondria that play the opposite effects in mitochondrial function, as an increase of ATP production will promote cell survival and enhance cardiac functions, an increase of mPTP opening will trigger cell death and impair the cardiac function. Studies have shown that mitochondrial Ca^2+^ uptake is involved in these two important mitochondrial functions. For example, under physiological conditions, an increase in mitochondrial Ca^2+^ uptake will enhance the ATP production, which will be beneficial for the cardiomyocyte function. However, in pathological conditions, an increased mitochondrial Ca^2+^ uptake will induce mitochondrial Ca^2+^ overload, triggering mPTP opening and resulting in deleterious effects in the cardiomyocytes [5].

### 3.1. Mitochondrial Ca^2+^ Uptake and ATP Production

The heart is a high energy demanding organ. Mitochondria occupy a large portion of each myocyte to supply about 90% of the cardiac energy requirement [31,32]. It must respond fast to meet the rapid change in energy-demands from rest pace to exercise pace. Ca^2+^ flux has been identified as the messenger to regulate ATP production. In the 1980s, Denton and McCormack found that Ca^2+^ can activate the key TCA cycle enzymes: pyruvate dehydrogenase (PDH), oxoglutarate dehydrogenase (OGDH), and can isocitrate dehydrogenase (ICDH) at a micromolar level (within physiological range) [33]. The activation of those dehydrogenases leads to an increase of NADH oxidation, resulting in an increase of ATP production. There are other mechanisms involving Ca^2+^-stimulated ATP production by stimulating ATPase. For example, Ca^2+^ stimulates ATP production by binding to ATPase inhibitor protein (IF1) or ATPase (complex V) itself [34]. This concept was further supported by the studies in other human cell lines: HEK-293T and HeLa cells, by Dr. Docampo’s group. They found that mitochondrial Ca^2+^ regulates ATP production through MCU interacting with a subunit c of ATP synthase [35].

In addition to the interaction between Ca^2+^ with ATP synthase in mitochondria, recent studies also found a dose-dependent association between Ca^2+^ concentration and ATP production, which may link to the mitochondrial membrane potential (ΔΨ). Using skeletal muscle mitochondria, Sivitz’s group found that the mitochondrial Ca^2+^ stimulates ATP production at low concentrations (the nanomolar level) while inhibiting the process at a high concentration (the micromolar level) [36]. They found that, while a low concentration of Ca^2+^ induced oxidation of NADH which was dependent on complex I substrate: glutamate/malate at a certain respiratory status, the inhibitory effect of high Ca^2+^ changed ΔΨ independent of the opening of mPTP, suggesting a potential alternative pathway in ΔΨ. In addition, by using adult rats’ cardiomyocytes, Boyman’s group’s results suggested a pathway regulating voltage-dependent ATP production, through which ATP production was regulated by the Ca^2+^ flux and induced changes in IMM potential as well as the modulation of pyruvate and glutamate dehydrogenase activity by matrix Ca^2+^ [37]. This regulation is independent of ATP synthase or other electron transport chain complexes. The authors observed similar regulation in skeletal muscles as well. Notably, all of the findings are from isolated mitochondria or isolated cells, which may differ from the actual in vivo conditions due to the technological limitations. The results may also vary due to the use of different cell types and the method of mitochondrial isolation or measurements.

### 3.2. Mitochondrial Ca^2+^ and mPTP Opening

The mPTP is a transmembrane protein complex and is located in the IMM. The pore opening can release a substance as big as 1.5 kDa, and it usually opens in response to the matrix Ca^2+^ overload. Therefore, it plays a crucial role in the pathological condition, although the molecular structure and regulating proteins are still under investigation. Current reports are debating, for example, while some researches indicate that VDAC at OMM and adenine nucleotide translocase (ANT) at IMM are the structural members of mPTP [38,39], the studies with a VDAC knock-out model and an ANT inhibitor suggested that those molecules could be dispensable [12,40,41,42]. In addition, it was widely accepted that cyclophilin D was a major regulator of mPTP, whose activation stimulates pore opening, and its inhibition has been shown to reduce ischemia–reperfusion injury in various animal models as well as in humans by the inhibition of mPTP [43,44,45]. However, a large clinical trial conducted by international groups demonstrated that cyclosporine A, a cyclophilin D inhibitor, failed to protect the heart in patients with myocardial infarction [46]. Moreover, phosphate carrier (PiC) and F1F0ATP synthase at IMM are found to be involved in the regulation of pore opening [47]. Recently, two independent studies using purified F1F0ATP synthase from dog or pig heart respectively, proved more evidence on the essentialness of this enzyme on mPTP activity/structure. It has been shown that, under the Ca^2+^ treatment, purified ATP synthase activity can be regulated by an mPTP specific agonist and inhibitor, and the purified monomer forms are in voltage-gated and Ca^2+^-activated channels [48,49]. A study points out that the sensitivity of pore opening varies according to the respiratory status and substrates availability [50]. Mitochondria isolated from rats’ liver showed that the pore opening is less sensitive towards ROS, while it is more sensitive to complex substrates succinate. This finding is interesting as it indicates that the metabolic status may change the sensitivity of pore opening. This finding may have an important impact on the organs with high metabolic rates, such as the heart.

Despite the uncertain regulating mechanisms, it is commonly accepted that a high concentration of matrix Ca^2+^ concentration is one of the factors triggering the pore opening. The consequence of pore opening is a failure in maintaining Ca^2+^ homeostasis, losing ΔΨ, the rapture of mitochondria, and cell death. Although mPTP opening was considered an alternative way to release matrix Ca^2+^ and reduce the potentially harmful effect of Ca^2+^ overload, thus, may play a protective role, the later on *ex vivo* experiments cannot support this hypothesis [5]. More evidence suggest that the modulation of mPTP opening is becoming an effective strategy to protect cell death from stress.

## 4. The Molecular Basis and Regulation of the Mitochondrial Ca^2+^ Uptake

It has been shown that mitochondrial Ca^2+^ uptake is regulated by a multi-protein complex including the main pore unit, MCU and MCU regulatory unit b (MICUb), and its key associate proteins, such as mitochondrial Ca^2+^ uptake proteins (MICU1, MICU2, and MICU3), and essential MCU regulator (EMRE).(as shown in Figure 2). This complex cooperates tightly to regulate the mitochondrial Ca^2+^ uptake in different tissues. The MICU family located in the IMS, while MCU, EMRE and MCUb are transmembrane proteins [7]. The expression of the associated proteins varies among the tissues [51] due to the different functions and energy demands of the tissues. In the heart, the MCUb expression level is relatively higher than in the other tissues, and it is proposed to be a negative regulator of MCU, but the mechanism is unknown [52]. It was also reported that MCU regulator 1 (MCUR1) bound to MCU and EMRE and proposed to affect MCU activity as well as uniporter complex assembly [53]. EMRE also binds to MCU to stabilize the opening pore during the Ca^2+^ transition [54]. However, the mechanisms of the interaction and regulation among these proteins are still largely unknown and are being investigated. We summarized here the updated information related to this complex, in terms of the molecular basis and regulatory mechanisms, focusing on the MCU and three members of MICUs (MICU1, MICU2, and MICU3) since they are relatively well-studied among the MCU complex-associated proteins. Furthermore, under pathological conditions, mPTP can act as a backup Ca^2+^ efflux mechanism for Ca^2+^ uptake, and maintain Ca^2+^ homeostasis. For example, Saotome’s group identified in rat cardiomyocytes, dissipation of ΔΨ would open mPTP and allow Ca^2+^ influx into mitochondria [55,56].

### 4.1. The Structure and Biological Function of MCU and Its Role in the Cardiac Function

MCU presents in almost all cells and is a highly selectively calcium channel responsible for Ca^2+^ uptake in mitochondria [57]. There are two transmembrane domains in MCU locating in IMM, and both N- and C-terminal domains stay inside the matrix, while the N-terminal domain (NTD) is believed to regulate Ca^2+^ uptake rate [58]. Although the studies on the molecular nature of the MCU complex began in the 1970s, it is until 2011 when MCU was identified and confirmed as an essential component of the mitochondrial Ca^2+^ uniporter, based on two representative papers published at the same time on Nature [59,60,61]. Since then, the structure has been elucidated in various species, including *fungi, C. elegans, zebrafish*, and humans [57]. The transmembrane and NTD structures are preserved among species, indicating a critical role for mitochondrial Ca^2+^ homeostasis. For example, a study in *C. elegans* showed that there is a conserved sequence Asp240-x-x-Glu243(DXXE) in MCU, which is critical in pore opening and selectivity filters [62]. In humans, a similar motif is D261-X-X-E264, and the mutation of D or E would lose MCU activity completely [53]. Although MCU’s posttranslational regulation mechanisms remain unclear, Dong’s group has identified that a cysteine in N-terminus can be oxidized by ROS, which results in the activation of MCU to enhance Ca^2+^ uptake, leading to mitochondrial Ca^2+^ overload and cell death [63]. This study indicates another pathway of oxidative stress-induced Ca^2+^ overload besides affecting the structure of mPTP.

Since mitochondrial Ca^2+^ overload-induced cell death is critical to the development and progress of heart failure, and MCU plays an essential role in mitochondrial Ca^2+^ uptake, researches have been conducted in investigating the impact of MCU in cardiac dysfunction. At least three mouse models with the MCU disruption have been developed by different groups [64], including a systematic MCU knock-out (KO) [65], an inducible cardiac-specific MCU KO (cMCU KO) [66], and a cardiac-specific transgenic mouse with overexpression of a dominant-negative form of MCU (DN-MCU) [67]. However, the results among these models remain contradictory. For example, MCU’s cardiac-specific deletion in an cMCU KO mouse reduced cell death under I/R injury, which may occur due to the less mitochondrial Ca^2+^ overload under stress. However, this protective effect was not observed in MCU KO and DN-MCU mice’s hearts [65,66,68]. In addition, it has been shown that, compared to the WT mice, there is no significant difference in basal mitochondrial metabolism in the mitochondria isolated from the cMCU KO mouse. However, it cannot rapidly uptake enough Ca^2+^ and adjust the energy supply in a Ca^2+^-dependent manner under stimulation. Interestingly, the mitochondrial Ca^2+^ concentration in cMCU KO mice is lower than WT at the beginning of catecholamine treatment; it slowly catches up to the level of WT mice. These findings suggested that MCU plays a vital role in the rapid response towards stimulation. Moreover, as both DN-MCU and cMCU KO mice are insensitive towards catecholamine challenges, MCU KO mice act normally [57,67,69,70,71]. Even though an MCU mouse model’s overexpression was never developed, an *in vitro* study showed that overexpressed MCU in mouse primary cortical neurons resulted in more cell death with Ca^2+^ overload [72]. WT hearts treated with an mPTP inhibitor showed a protective effect against I/R injury, indicating the key role of mPTP in I/R induced cell damage. This effect was not observed in the MCU KO and DN-MCU mice. This difference may imply that the interruption of MCU function at the early stage of the development may induce an alternative mechanism to attenuate the mPTP opening under stress. Studies have suggested that MCU involves developing and progressing cardiac dysfunction under stress; however, the underlying mechanisms remain largely unknown.

### 4.2. Mitochondrial Calcium Uptake Proteins (MICU1, MICU2, and MICU3)

MICU family members are key regulators of MCU and are relatively well studied compared with the other associate proteins. There are three paralogous (MICU1, MICU2, and MICU3) in mice and humans, and the expression level for each of them varies in different tissues [34]. It is known that MICU2 is dominantly expressed in visceral organs, and MICU3 can be found in skeletal muscle and nerve tissue [73]. MICU1 can form a heterodimer with MICU2 or MICU3, but this structure does not happen between MICU2 and MICU3. Researchers recently found that the absence of MICU1 in MCU complex results in other ions besides Ca^2+^ pass MCU pore, indicating that MICU1 is essential for the Ca^2+^ selectivity in mitochondria [74]. Evolutionarily, MICU1 existed earlier than MICU2 and MICU3 [75], resulting in a more complicated cellular function. A recent structure study indicated that in a low Ca^2+^ condition, MICU1 and MICU2 form a heterodimer, and that Glu242 in MICU1 and Arg352 in MICU2 are critical for this bonding. After Ca^2+^ binds to MICU1 and MICU2, MICU1 and MICU2 bind more tightly and open the MCU pore, and this interaction requires Phe383 in MICU1 and Glu196 in MICU2 [76]. Petrungaro et al. characterized the interactome of the human mitochondrial oxidoreductase Mia40 with MICU1, and showed that Mia40 primes MICU1 for heterodimerization with MICU2, which associates with MCU in a Ca^2+^-dependent manner. They found that changes in Ca^2+^ levels affected the interaction of MICU1 and MICU2 with MCU. Specifically, they showed that, at 0.7 to 2 mM Ca^2+^, the MICU1 dimer showed a reduced binding to MCU, while at concentrations between 1 and 3 mM, the rate of mitochondrial Ca^2+^ uptake was the highest [77]. In addition, in a cell line model, Jessica Matesanz-Isabel et al. also showed that MICU2 behaves as a pure inhibitor of MCU at a low cytosolic Ca^2+^ level, its effects decreased as cytosolic Ca^2+^ increased and disappeared above 7 μM. On the other hand, MICU1 has a double role in MCU regulation and is inhibitory at a low Ca^2+^ level but is activated when the Ca^2+^ level is more than 2.5 μM [78] (Figure 2). The results suggest that MICU1 acts as a lock and selective gate of MCU, while MICU2 and MICU3 may act as a regulator of MICU1. This statement was supported by MICU deletional mouse models. MICU1 KO mice show increasing Ca^2+^ uptake, mitochondrial dysfunction, and impaired energy production at baseline [79]. However, the MICU2 KO mouse showed abnormal cardiac relaxation, but no Ca^2+^ overload was observed. Instead, the Ca^2+^ uptake is relatively slow in the MICU2 KO mouse [80].

Ca^2+^ activates MICUs; therefore, all three paralogous share similar Ca^2+^ binding sites and EF hands motif, while the EF-hands help MICU1 detect Ca^2+^ concentration and open MCU. Recently, the human MICU crystal structure has been resolved using EM [81]. They proposed that a MICU1-MICU2 heterodimer covers MCU pores at rest using a five-residue K/R ring. When Ca^2+^ concentration increases (over 1uM), the conformational change of the complex weakens the interaction between MICU1-MCU and opens the MCU pore [82]. This finding suggested an alternative regulation mechanism from a previous study. The previous study suggested that IMS resident oxidoreductase Mia40 links the MICU dimer to MCU by a disulfide covalent bond. This bonding leads to binding between MICU1 and MCU at a low Ca^2+^ concentration and dissociates when the Ca^2+^ concentration is high. The disulfide bond has also been reported between the MICU1-MICU3 heterodimer and MCU in primary cortical neurons [83].

Moreover, it has been found that the protein expression of MICU1 and MICU2 can affect MCU expression. Studies showed that knockdown of either MICU1 or MICU2 could decrease MCU expression [7]. It may suggest a molecular mechanism in regulating MCU expression in different tissue types to match the specific energy requirements. There are limited studies on MICU in cardiovascular disease. It has been reported that MICU1 can preserve cardiac function and prevent cell death in mice with diabetes-induced cardiomyopathy [84].

## 5. Other Regulators in Mitochondrial Ca^2+^ Uptake

Although the molecular mechanisms underlying the regulation of mitochondrial Ca^2+^ are not fully understood, a few new regulators have been discovered. For example, a study showed that ROS might involve in the regulations of ATP production via the mitochondrial Ca^2+^ uptake [10]; SLC25A23 (an Mg-ATP/Pi solute carrier) has been purposed to regulate MCU complex [85], and ER has an impact on mitochondrial Ca^2+^ uptake by interacting with mitochondria [86].

In addition, a recent study has shown that an ATPase protein named valosin-containing protein (VCP) is involved in the regulation of mitochondrial Ca^2+^ uptake [13]. VCP belongs to an ATPase associated family with four structural domains [87]: an N-terminal substrate-binding domain, two ATPase domains, and a C terminal domain. It has been shown that VCP is involved in multiple cellular pathways in many tissues such as autophagy, apoptosis, and ubiquitin degradation [88]. Mutation of VCP would result in energy impairment in cells [89]. Recently, VCP has been found to play an essential role in mitochondrial function, including mitochondrial respiration, ATP production and mPTP opening. Overexpression of the VCP protects cardiomyocyte against stimulated death and reduces the I/R induced infarct size in the heart [13,90,91,92]. Notably, the most recent study found that VCP protects the heart from the stress by preventing excessive Ca^2+^ overload through inhibiting Ca^2+^ uptake [13]. The study also showed that the VCP-mediated inhibition in Ca^2+^ uptake attenuates the stress-induced reduction of ATP production and the mPTP opening [13]. The mechanism of this protective effect was linked to the VCP-mediated selective degradation of MICU1, which subsequently reduced the uptake of Ca^2+^ under stress [13]. These studies identified VCP as a new regulator of mitochondrial Ca^2+^ in the heart; it also provides evidence supporting a new mechanism of the dual role of VCP-mediated Ca^2+^ uptake in the ATP production and the mPTP opening.

## 6. Conclusions and Future Direction

As summarized in Figure 1, mitochondrial Ca^2+^ uptake is a double-edged sword regulating cardiac function at both physiological and pathological conditions. In the heart, mitochondrial Ca^2+^ is essential to maintain contractility, rapidly stimulating ATP production, while mitochondrial Ca^2+^ overload would induce mPTP opening to lead to cardiac damage and dysfunction. Therefore, maintaining calcium homeostasis in the mitochondria becomes very important in preventing heart disease, as representative studies have shown in Table 1. Although many interesting findings are obtained, investigations need to be conducted to better understand the underlying mechanism of the regulation in mitochondrial calcium homeostasis, to improve the detecting techniques and to determine the real-time dynamic alteration of calcium inside the cells, and to develop the therapeutic strategy to control the harmful effects caused by Ca^2+^ overload.

One of the potential research interests could be discovering the molecular basis and regulatory mechanism of the mitochondrial homeostasis. First, despite the progress on the studies of the biological function of the MCU/MICUs complex, it is far from fully understood. The current studies have a lot of conflict results and unexplained mechanisms. Further exploring the structural and function and the interaction among these structural proteins will help determine the regulatory control within this complex. Aside from these known proteins, attention should be paid to other potential associated proteins for which the functions are poorly understood, such as EMRE, MCUb, and other unknown proteins. Mainly, the different expressions of MICU isoforms in various tissues, indicating that a potential tissue-specific regulation of mitochondrial Ca^2+^, has not been studied. Third, it is also essential to determine the regulation in controlling the balance between the calcium uptake and release systems, such as the cellular regulation and interaction among those components, like the structure of the Ca^2+^ exporter NCX.

Another important research area is to find out the distinct regulatory mechanisms in the physiological and pathological conditions. There might also be a different mechanism involved in the development and the adult cardiac function. In addition, the link between the mitochondrial Ca^2+^ uptake with other mitochondrial functions, such as ATP production, mPTP opening, and respiration needs to be further explored. Furthermore, more new regulators involving in these mitochondrial functions, like VCP, and their mechanisms need to be discovered in future studies.

In addition, as we mentioned above, most of our research results were observed from the *in vitro* studies from the cells or isolated mitochondria. These measurements could be different from the physiological condition that occurred *in vivo*. The detecting techniques need to be developed to better reveal the real-time alterations of the calcium dynamics *in vivo*.

Finally, based on the animal models and *in vitro* studies, therapeutic drugs need to be identified by targeting mitochondrial Ca^2+^ homeostasis regulation. An mPTP inhibitor has been developed and applied with clinical promise. New therapeutic strategies need to be explored in the future.

## Figures and Tables

**Figure 1 ijms-21-07689-f001:**
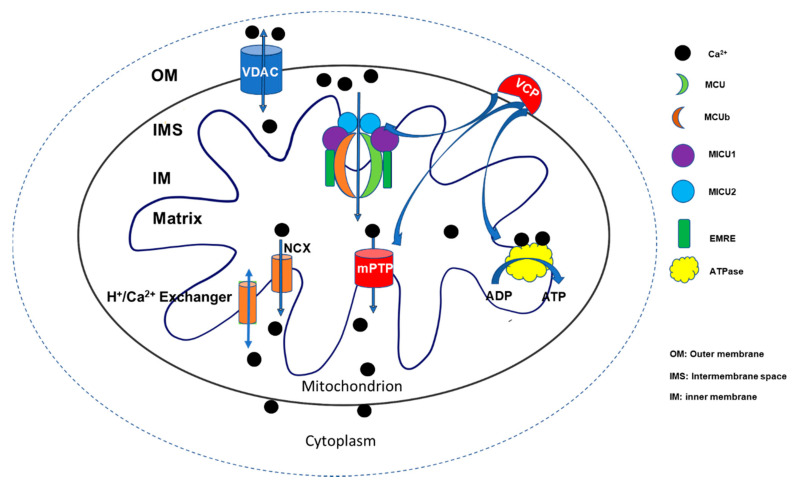
Summary of the regulation in mitochondrial Ca^2+^ homeostasis. Two major regulatory systems participate in regulating mitochondrial Ca^2+^ homeostasis by releasing Ca^2+^ into the cytosol or up-taking Ca^2+^ from the cytosol. Ca^2+^ enters the mitochondrial intermembrane space (IMS) via voltage-dependent anion channel 1 (VDAC1) and is then transferred into the mitochondrial matrix through a mitochondrial Ca^2+^ uniporter (MCU) by interacting with a variety of regulatory proteins, such as calcium uptake proteins (MICUs), as well as essential MCU regulators (EMRE) to constitute a comprehensive functional complex that controls the uptake of Ca^2+^. On the other hand, the Ca^2+^ stored in the matrix can be pumped back to IMS through Na^+^/Ca^2+^ (NCX) or H^+^/Ca^2+^ exchangers and mitochondrial permeability transition pore (mPTP) openings. Ca^2+^ participates in multiple energy generation processes, such as the activation of ATPase, to stimulate ATP production. Valosin-containing protein (VCP) represents a novel regulator involving the control of Ca^2+^ uptake, ATP production, and mPTP opening [7,14].

**Figure 2 ijms-21-07689-f002:**
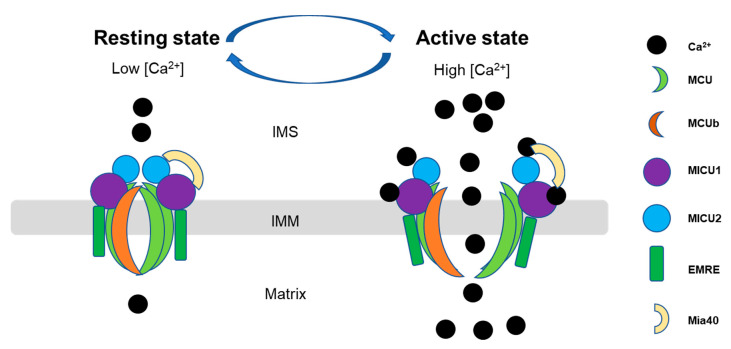
Scheme of the dynamic function of the mitochondrial calcium uniporter (MCU) complex. Mitochondrial Ca^2+^ uptake is controlled by a multiprotein complex which locates as a transmembrane pore in the inner membrane, consisting of the pore-forming subunits (MCU and MCUb), structural protein (the essential mitochondrial Ca^2+^ uniporter regulator (EMRE)) and Ca^2+^ sensitive proteins (the mitochondrial Ca^2+^ uptake (MICU)). MCU- complex mediated Ca^2+^ entry into mitochondria in a Ca^2+^-dependent manner: at low Ca^2+^ in the resting state, Mia40 introduces an intermolecular disulfide bond that links MICU1 and MICU2 in a heterodimer, which binds to MCU to ensure MCU gatekeeper activity, preventing mitochondrial Ca^2+^ from entering into the mitochondrial matrix (**left**). Upon an increase in Ca^2+^ concentrations in the IMS, Ca^2+^ binding to the MCIU1-MICU2 dimer leads to structural changes that lower the affinity toward MCU, dissociates MICU proteins from MCU, and actives MCU channel activity, allowing efficient mitochondrial Ca^2+^ uptake (**right**) [51,77]. IMS, intermembrane space; IMM, inner mitochondrial membrane.

**Table 1 ijms-21-07689-t001:** Summary of mitochondrial Ca2+ uptake studies related to cardiac activity.

Molecule	Model	Outcome	Ref
ATPase	Mitochondria isolated from pig heart	Ca^2+^ activates ATPase	[4,93]
	Rats cardiomyocytes	Cardiac heavy workload increases matrix Ca^2+^ concentration to meet the high energy requirement.	[20,94]
MCU	Systematic MCU KO mice	No protection against I/R injury; Acting normal under catecholamine stimulation	[65,69]
	Inducible cardiac-specific MCU KO mice	Reduced cell death under I/R injury Insensitive towards catecholamine stimulation	[66,70,71]
	Cardiac-specific dominant-negative MCU overexpression mice	No protection against I/R injury Normal heart rate at rest, but unable to accelerate heart rate with catecholamine stimulation	[67,68]
mPTP	Multiple models	Inhibition of mPTP protect heart from I/R injury and ROS damage	[21,38,95]
MICU1	MICU1 KO mice	Increasing Ca^2+^ uptake, mitochondrial dysfunction, and impaired energy production at baseline	[79]
	db/db mice	Overexpression of MICU1 preserves cardiac function in diabetic db/db mice	[84]
MICU2	MICU2 KO mice	Abnormal in cardiac relaxation	[80]
NCX	Dog cardiomyocytes	Increase in cytosolic Na^+^ levels during heart failure causes mitochondrial Ca^2+^ transport out through NCX, and impaired Krebs cycles	[29]
VCP	Cardiac specific VCP overexpression mice	VCP protects the heart from the stress by preventing excessive Ca^2+^ overload through inhibiting Ca^2+^ uptake	[13]

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
