# Peer review of "The Physiological and Pathological Roles of Mitochondrial Calcium Uptake in Heart"

_ijms, 2020, doi:10.3390/ijms21207689_

Round 1

Reviewer 1 Report

The review describes the calcium dynamics in the heart’s cells, focusing the attention on molecular mechanisms that either improve or reduce [Ca2+] in mitochondria. Since the title of the review indicates “physiological and pathological conditions”, the readers are waiting for more attention on pathological events induced by Calcium alterations, providing connections with specific pathologies related to these events (e.g. ROS burst in transplanted heart, however authors might pinpoint to other pathologies). For example, authors might include some sentences concerning animal models and/or patient’s stories in which calcium dynamics are affected. Alternatively, authors might include a new section, discussing the macroscopic effects of altered calcium dynamics in animal models.

The overall level of the document is moderate and some corrections should be introduced. Some comments (listed below) may improve the review.

Line 11 and 12

Redundant content

The sentence is too similar to the title.

Line 28

Reformulate the sentence.

  • Mitochondria are the main powerhouses in the heart cells (do not generalize the concept).

  • generate ATP is improper. Use re-supply the cells with ATP or ... catalyze the regeneration of ATP…

Line 34-38

Reformulate the sentences.

The meaning is unclear

Line 51

Unspecific concept:

Ca2+ is kept at a low level. (include nM reference or replace with [Ca2+] is constantly controlled through two…)

Line 119

Major issue

Where is the calcium if not in SR or in the cytosol? If SR calcium reload is blocked Ca2+ should increase in the cytosol, in general.

Anyway, another sentence is required to justify the text. To specify if calcium is either extruded in ECM or excluded by the binding with proteins, or the mechanism is currently unknown.

Line 130.

Function is a Plural word.

In genetics, the function is the purpose for which a gene is employed by the evolution.

Since mitochondria have some purposes in the cells, this means the mitochondria have some different functions, not a single one.

Line 134

Redundant content

it is considered a double-edged sword in regulating mitochondrial function

Line 167

Italic

In vivo should be texted with an italic font.

Line 173

Major issue

mPTP molecular identity is still debated. Since ANT and VDAC are dispensable components of mPTP. They cannot be described as core mPTP proteins and ATP synthase cannot be dismissed as “involved in the regulation of pore openings”. Some recent papers may shed light on this topic and should be included in the discussion.

>Purified F-ATP synthase forms a Ca 2+-dependent high-conductance channel matching the mitochondrial permeability transition pore. Andrea Urbani et al Nat Commun. 2019 Sep 25;10(1):4341. doi:10.1038/s41467-019-12331-1.

>A mitochondrial mega-channel resides in monomeric F(1)F(O) ATP synthase. Mnatsakanyan N et al. Nat Commun. 2019 Dec 20;10(1):5823. doi: 10.1038/s41467-019-13766-2.

Line 265

Unspecific concept

Include nM reference. I would like to see also the cellular model

Line 343

Italic

In vitro should be texted with an italic font

Line 400

Wrong Reference 

Ref 11, Names and surnames are swapped

Reviewer 2 Report

Concerning the submitted article « The physiological and pathological roles of mitochondrial calcium uptake in heart » by Lo Lai and Hongyu Qiu, I do not see any raison not to publish as it come, apart a small remark about the illustration. If the illustration has been inspirated by several other ones in diverse articles, it will be nice to have the citation of them in the figure legend.

I feel also that i twill be nice to have a design (in an other figure) of the mitochondrial Ca2+ uniporter (+ its regulatory associated protein) in a resting state and when the pore is active. See rizzuto for such a purpose…

And please have a look to the following references… That may be introduced into your reference list.

Spatial Separation of Mitochondrial Calcium Uptake and Extrusion for Energy-Efficient Mitochondrial Calcium Signaling in the Heart.

De La Fuente S, Lambert JP, Nichtova Z, Fernandez Sanz C, Elrod JW, Sheu SS, Csordás G. Cell Rep. 2018 Sep 18;24(12):3099-3107.e4. doi: 10.1016/j.celrep.2018.08.040.

Mitochondrial energetics and calcium coupling in the heart.

Kohlhaas M, Nickel AG, Maack C. J Physiol. 2017 Jun 15;595(12):3753-3763. doi: 10.1113/JP273609. Epub 2017 Mar 10.

Calcium at the Center of Cell Signaling: Interplay between Endoplasmic Reticulum, Mitochondria, and Lysosomes.

Raffaello A, Mammucari C, Gherardi G, Rizzuto R. Trends Biochem Sci. 2016 Dec;41(12):1035-1049. doi: 10.1016/j.tibs.2016.09.001. Epub 2016 Sep 28.

Round 2

Reviewer 1 Report

The authors significantly improved the quality of the review. The new version of the manuscript is fully acceptable because the authors fulfilled all the questions and the issues.

Author Response

Thanks for the reviewer for your time and your positive comments!